# Silicon as a Functional Meat Ingredient Improves Jejunal and Hepatic Cholesterol Homeostasis in a Late-Stage Type 2 Diabetes Mellitus Rat Model

**DOI:** 10.3390/foods13121794

**Published:** 2024-06-07

**Authors:** Marina Hernández-Martín, Alba Garcimartín, Aránzazu Bocanegra, Rocío Redondo-Castillejo, Claudia Quevedo-Torremocha, Adrián Macho-González, Rosa Ana García Fernández, Sara Bastida, Juana Benedí, Francisco José Sánchez-Muniz, María Elvira López-Oliva

**Affiliations:** 1Departmental Section of Physiology, Pharmacy School, Complutense University of Madrid, 28040 Madrid, Spain; marinh04@ucm.es; 2Pharmacology, Pharmacognosy and Botany Department, Pharmacy School, Complutense University of Madrid, 28040 Madrid, Spain; a.garcimartin@ucm.es (A.G.); roredond@ucm.es (R.R.-C.); clauquev@ucm.es (C.Q.-T.); jbenedi@ucm.es (J.B.); 3Nutrition and Food Science Department, Pharmacy School, Complutense University of Madrid, 28040 Madrid, Spain; amacho@ucm.es (A.M.-G.); sbastida@ucm.es (S.B.); frasan@ucm.es (F.J.S.-M.); 4Animal Medicine and Surgery Department, Veterinary School, Complutense University of Madrid, 28040 Madrid, Spain; ragarcia@ucm.es

**Keywords:** diabetic dyslipidemia, cholesterol homeostasis, silicon, functional meat food, liver, jejunum

## Abstract

Silicon included in a restructured meat (RM) matrix (Si-RM) as a functional ingredient has been demonstrated to be a potential bioactive antidiabetic compound. However, the jejunal and hepatic molecular mechanisms by which Si-RM exerts its cholesterol-lowering effects remain unclear. Male Wistar rats fed an RM included in a high-saturated-fat high-cholesterol diet (HSFHCD) combined with a low dose of streptozotocin plus nicotinamide injection were used as late-stage type 2 diabetes mellitus (T2DM) model. Si-RM was included into the HSFHCD as a functional food. An early-stage TD2M group fed a high-saturated-fat diet (HSFD) was taken as reference. Si-RM inhibited the hepatic and intestinal microsomal triglyceride transfer protein (MTP) reducing the apoB-containing lipoprotein assembly and cholesterol absorption. Upregulation of liver X receptor (LXRα/β) by Si-RM turned in a higher low-density lipoprotein receptor (LDLr) and ATP-binding cassette transporters (ABCG5/8, ABCA1) promoting jejunal cholesterol efflux and transintestinal cholesterol excretion (TICE), and facilitating partially reverse cholesterol transport (RCT). Si-RM decreased the jejunal absorptive area and improved mucosal barrier integrity. Consequently, plasma triglycerides and cholesterol levels decreased, as well as the formation of atherogenic lipoprotein particles. Si-RM mitigated the dyslipidemia associated with late-stage T2DM by Improving cholesterol homeostasis. Silicon could be used as an effective nutritional approach in diabetic dyslipidemia management.

## 1. Introduction

Type 2 diabetes mellitus (T2DM) is a major public health problem worldwide. It is well known that T2DM often coexists with lipid metabolism disorders, developing into severe dyslipidemia [1]. T2DM patients often present elevated levels of triglycerides, small low-density lipoprotein (LDL) particles, as well as low levels of high-density lipoprotein cholesterol (HDLc), which is strongly associated with an increased cardiovascular (CV) risk [2]. Therefore, T2DM treatment should also be aimed at controlling dyslipidemia and maintaining cholesterol homeostasis as an important therapeutic goal. Among others, Diabetes guidelines [3] have recommended nutritional and pharmacological treatments to target LDLc [3,4,5]. However, a significant portion (44–67%) of patients with T2DM do not reach the recommended treatment goals for lowering LDL cholesterol, indicating a need for new cholesterol-lowering therapies. In this context, functional foods can complement pharmacotherapy to enhance lipid-lowering effects, potentially enabling a reduction in medication dosage [6,7,8,9]. It has been reported that cholesterol-lowering functional foods may affect intestinal cholesterol absorption by regulating the genes of cholesterol transporters/proteins such as Niemann-Pick C1-Like-1 (NPC1L1), acyl-Coenzyme A cholesterol acyltransferase-2 (ACAT2), microsomal triglyceride transfer protein (MTP), ATP-binding cassette subfamily G members 5 and 8 (ABCG5/8) transporters, and the liver X receptor (LXRα/β) transcription factor which controls the transcription of absorption and efflux of cholesterol transporter genes [10]. Also, the inhibition of hepatic 3-hydroxy-3-methylglutaryl CoA reductase (HMGCR) acting as a rate-limiting enzyme in cholesterol synthesis, hepatic MTP activating excretion of very-low-density lipoprotein cholesterol (VLDLc), and the upregulation of low-density lipoprotein receptor (LDLr) responsible for LDLc removal from plasma have been shown as possible mechanisms of hypocholesterolemic action of functional foods [7,11].

Studies have suggested that excessive consumption of red and/or processed meats with high amounts of cholesterol and saturated fatty acids associated with Western dietary patterns may have adverse effects on both plasma lipids and glucose levels, leading to an atherogenic profile in T2DM patients [12,13]. Although high meat consumption is not recommended for T2DM patients [14], its intake continues to be high. Indeed, meat consumption is widely accepted by consumers because meat is tasty. Additionally, it may be considered that meat is an important source of pro-health compounds [15], which are difficult to obtain in sufficient amounts from other sources [16]. Furthermore, meat is an excellent matrix for the introduction of different bioactive compounds to design functional foods [17,18]. The inclusion of functional ingredients can modify the bio-accessibility and bioavailability of components such as fat, making meat a potentially healthier product as an effective tool to prevent and treat diabetic dyslipidemia [19].

Silicon is a nutrient whose physiological benefits have been the subject of extensive research in recent years. Its effects in reducing the risk of atherosclerosis by partially blocking dyslipidemia in health and illness have been demonstrated [20,21]. Restructured meat (RM) enriched with silicon (Si-RM) has been proposed as a functional food aimed at improving lipoprotein composition [22]. Si-RM has demonstrated hypoglycemic and hypolipemic properties, improving the atherogenic lipid profile, VLDLc oxidation, and LDLr gene (*Ldlr*) expression in aged early-stage T2DM rats [23,24,25]. Si-RM could be an alternative food for dyslipidemic T2DM patients in order to maintain HDLc and LDLc levels. The duodenal molecular mechanisms inhibiting cholesterol absorption have already been described [26], but the involvement of Si-RM’s hypocholesterolemic effect on cholesterol metabolism has not been studied in depth.

Due to plasma cholesterol homeostasis being a process primarily mediated by lipids transport to and from the liver and intestine, and because the jejunum is the main site of intestinal absorption, this study aims to examine the impact of Si-RM on cholesterol absorption and metabolism markers in a late-stage T2DM rat model. Concretely, the effects of Si-RM on the cholesterol absorption process have been studied by regulating the expression of intestinal cholesterol transporters/enzymes such as NPC1L1, ACAT2, and MTP, the measuring of the jejunal absorptive area, and the renewal and integration of the intestinal barrier. Additionally, cholesterol efflux by ABCG5/G8, ABCA1, and the transintestinal cholesterol excretion (TICE) pathway were activated by LXRα/β; hepatic cholesterol biosynthesis worked by activation of HMGCR; also, hepatic formation and secretion of VLDL by MTP had been evaluated.

## 2. Materials and Methods

### 2.1. Diets, Animal Model, and Experimental Design

Restructured meat matrix (RM) was prepared according to Schultz-Moreira et al. [27]. Details of the diet formulation and composition are in Appendix A. RM was processed from lean minced meat, consisting of an equal mix of pork and veal (50% each) from a local store (reference SF302; Safe Custom Diet, S.L., Augy, France). This mixture was blended with lard for 1 min using a grinder homogenizer connected to a cooling bath at 2 °C (Stephan Universal Machine UM5, Stephan u. Söhne GmbH and Co., Hameln, Germany). After preparation, the meat mixtures were freeze-dried in a LyoAlfa 10 freeze dryer (Telstar, Terrassa, Spain) for 48 h. The freeze-dried products were then ground into a fine, homogeneous powder using a refrigerated mincer (Stephan Universal Machine UM5, Stephan u. Söhne GmbH and Co., Hameln, Germany) for 2 min. Thus, each kilogram of diet consisted of 30% meat mixture and 70% purified formulated diet, which were mixed and sieved three times until a completely homogeneous powder was obtained. The silicon-enriched meat (Si-RM) was prepared in the same way as the control meat, but with the addition of silicon. The form of silicon used in the diet was choline-stabilized orthosilicic acid (H_4_SiO_4_) obtained from silicium organique G57TM (Glycan Group, Geneva, Switzerland), a major form of bioavailable silicon for both humans and animals. The silicon dose (2 mg/kg b.w./day) was based on the differences in silicon intake between Western and Eastern populations, as described by Garcimartin et al. [22]. To achieve the desired supplementation, a certain amount of organic silicon containing 67 mg of silicon was added to 1 kg of the meat mixture. This resulted in a final silicon concentration of 20 mg per kg of the overall diet, which is safe, as previous studies have demonstrated [25,26].

Twenty-four male Wistar rats, each 2 months old (Harlan S.L., Barcelona, Spain), were used in this study. The rats were randomly divided into three equal groups. After a 7-day acclimatization period, eight rats were fed a high-saturated-fat diet (HSFD) made of RM (50% pork/50% veal) for eight weeks to induce early-stage T2DM (ED). The other 16 rats were fed a high-saturated-fat–high-cholesterol diet (HSFHCD, HSFD with the addition of 1.4% cholesterol plus 0.2% cholic acid) based on RM for three weeks to induce late-stage T2DM (LD). Following this period, these rats received an intraperitoneal injection of streptozotocin (STZ, 65 mg/kg b.w.) and nicotinamide (NAD, 225 mg/kg b.w.) (both from Sigma Aldrich, Madrid, Spain). Four days later, fasting hyperglycemia was confirmed, and the animals were divided into two groups: the LD group continued the RM/HSFHCD, while the LD-Si group received Si-RM/HSFHCD, with a Si dose of 2 mg/kg b.w./day. The ED group represents early-stage T2DM characterized by insulin resistance and hyperglycemia with insulinemia, while the LD group mimics a later phase of T2DM, characterized by higher glucose concentrations and hypoinsulinemia [25,26]. Rats had ad libitum access to tap water and food. Daily feed intake and body weight were recorded, and cholesterol intake was calculated. During the final week, fecal excretion was measured daily. Rats were anesthetized with isoflurane (5% *v*/*v*) before euthanasia, and blood was collected from the descending aorta using a heparinized syringe. The jejunum and liver were dissected, weighed, and processed (Figure 1).

Animals were housed in pairs and kept at a controlled temperature (22.3 ± 1.9 °C) and light (12 h light/dark cycles) at the Center for Animal Experimentation of Alcalá University, Madrid, Spain (registration number ES280050001165). The experiments were conducted in accordance with Directive 2010/63/EU on the protection of animals used for scientific purposes. This study was approved by the Advisory Committee for Science and Technology of Spain (project AGL2014-53207-C2-2-R and PID2019-103872RB-I00 and /AEI/10.13039/501 100 011 033) and by the Ethics Committee of the Complutense University of Madrid, Madrid (Spain).

### 2.2. Plasma Glucose and Insulin Level Determinations

Plasma was isolated by centrifugation for 10 min at 986× *g*, and glycemia was quantified immediately in a plate reader (SPECTROstar Nano, BMG LABTECH, Offenburg, Germany) at 492 nm using the GOD kit (Spinreact, Barcelona, Spain). Insulin was measured only at the end using an ELISA kit (Rat insulin Elisa KIT, ELR-Insulin, RayBiotech, Inc., Peachtree Corners, GA, USA). The color intensity was evaluated at 450 nm using a microplate reader (SPECTRO star Nano BMG LABTECH, Offenburg, Germany). HOMA-IR was calculated as follows: [fasting insulin (μIU/mL) × fasting glucose (mmol/L)/22.5].

### 2.3. Plasma Lipoprotein Cholesterol Determinations

Isolation of lipoprotein fractions was carried out from ultracentrifugation of 2 mL of plasma in saline gradient for 21 h 40 min at 272,000× *g* (40,000 rpm) at 4 °C, according to Terpstra et al. [28], and modifications by Olivero-David et al. [29]. Ultra-clear tubes (Beckman, Palo Alto, CA, USA) labeled to 1mL volume and a SW-40.1 rotor (Beckman L8-70M, Palo Alto, CA, USA) were used for separation of lipoprotein fractions, as well as the conventional boundaries for rats of the different lipoprotein classes (VLDL, ρ_20_ < 1.0063 g/mL; intermediate density lipoprotein (IDL), ρ_20_ = 1.0063–1.019 mg/mL; LDL, ρ_20_ = 1.019–1.057 g/mL; HDL, ρ_20_ > 1.057 g/mL).

Total cholesterol, total TG, VLDLc, VLDLtg, IDLc, LDLc, and HDLc from isolated lipoprotein fractions were quantified using the colorimetric kits Triglyceride-LQ and Cholesterol-LQ (Spinreact, Barcelona, Spain). Their measurements were made in plate readers at 492 nm (SpectroStar Nano, BMG, LABTECH, Offenburg, Germany). The atherogenic index (AI) was determined as follows: non-HDLc/HDLc ratio.

### 2.4. Fecal Fat Analyses

Fecal fat extraction was performed following the gravimetric method based on the Bligh and Dyer chloroform/methanol total lipid extraction as described by Olivero-David et al. [30]. One gram of dried feces was hydrated with 2 mL of water for 12 h at 4 °C and homogenized in an Ultraturrax T25 (IKA) until a homogeneous solution was obtained. Four milliliters of a chloroform/methanol mixture (1:1) were added and homogenized again. After centrifugation for 10 min at 750× *g*, the chloroform phase was collected. This process was repeated three times. Subsequently, the solvents were evaporated with a rotary evaporator (Rotavapor R-200, Büchi, Barcelona, Spain) and the residue of fat was weighed.

### 2.5. Histological Procedure

Sections of jejunum and liver were fixed in 10% formaldehyde and embedded in paraffin, after which serial sections 4 µm-thick were prepared. Hematoxylin–Eosin (H&E), periodic acid Schiff (PAS) and Alcian Blue (AB) pH 2.5 staining were performed for histological analysis. Images were obtained under Leica DM LB2 light microscopy and with a Leica DFC 320 camera, (Leica, Madrid, Spain) and quantified with ImageJ software (Fiji Image J Software; v.1.52i, NIH, USA).

Villi height was measured from the tip to the base and the width was obtained at the villi base level from H&E sections. Intestinal crypt depth was measured as the distance from the top of the crypt to the muscularis mucosae layer. At least 20 well-aligned villi and crypts per rat were tested. Both villi height/width and villi height/crypt depth ratios were calculated. The absorptive surface area of the jejunal villi was estimated by the following formula: Area of absorption surface of the villi = 2π × (Villi width/2) × Villi height. The number of goblet cells per villus or crypt of PAS and AB sections was counted with ImageJ software (Fiji image J; 1.52i, NIH, USA).

### 2.6. Immunohistochemical Staining

Jejunum and liver sections from six rats per group were fixed in 4% paraformaldehyde in 0.1 M phosphate buffer, pH 7.4, and dehydrated and embedded in paraffin. Afterwards, the tissue sections were deparaffinized, and endogenous peroxidase was inactivated with 3% hydrogen peroxide. The following primary mouse antibodies were used to incubate the sections at 4 °C overnight: anti-ABCG5, anti-ABCG8, anti-ABCA1 (Biorbyt Ltd., Quimigen, Madrid, Spain), anti-NPC1L1, anti-ACAT2, anti-MTP, anti-LXRα/β, anti-LDLr, anti-HMGCR, anti-proliferating cell nuclear antigen (anti-PCNA), anti-Occludin, and anti-Claudin 1 (Santa Cruz Biotechnology Quimigen, Madrid, Spain). The sections were incubated with biotinylated secondary antibody appropriate and stained horseradish peroxidase conjugated with streptavidin-biotin, 3,3′-diaminobenzidine (DAB), and Harris hematoxylin (Sigma-Aldrich, Madrid, Spain). The intensity of immunostaining for each antibody was measured using ImageJ 1.5.4 software (U.S. National Institutes of Health, Bethesda, MD, USA). A total of 10 fields per section per rat (200× magnification for image analysis) were selected and analyzed. Protein levels were evaluated by their staining pattern: weak (1), moderate (2), diffuse (3), or intense (4), and expressed as immunoreactivity score (IRS). All slides were examined by two different researchers in a blind manner. The PCNA labeling index (PCNA-LI) was calculated according to the following formula:

PCNA-LI = number positive nuclei × 100/Total number of cells per hemicrypt.


### 2.7. Terminal Deoxynucleotidyl Transferase dUTP Nick End Labeling (TUNEL) Assay

Jejunal sections were deparaffinized, rehydrated, and permeabilized with proteinase K (20 μg/mL) for 15 min at 37 °C. After quenching endogenous peroxidase activity using 3% hydrogen peroxide, sections were incubated with equilibration buffer for 10 min and then the terminal deoxynucleotidyl transferase reaction mixture was added to all sections except the negative control and incubated at 37 °C for 1 h. The reaction was stopped with saline sodium citrate buffer for 15 min. Biotinylated nucleotides were detected by streptavidin-HRP (1:500) for 30 min at room temperature, and then incubated with DAB until color development (5–10 min). The sections were counterstained with Harris’ hematoxylin. The number of positive apoptotic nuclei per villi was counted. To quantify the TUNEL index as the number of apoptotic cells × 100/number total cells/crypt height (%), we examined and counted at least 50 well-oriented perpendicular crypts for each animal at a magnification of ×400 using a Leica DM LB2 microscope equipped with a Leica DFC 320 digital camera (Leica Microsystems, Wetzlar, Germany).

### 2.8. Western-Blot Analysis

Jejunum and liver sections were homogenized with lyses buffer, and equal amounts of proteins (DS protein assays, Bio-Rad Laboratories, S.A., Madrid, Spain) were separated in denaturing SDS 10% or 15% polyacrylamide gels. Membranes were incubated overnight at 4 °C with the following primary antibodies: anti-ABCG5, anti-ABCG8 (Biorbyt Ltd., Quimigen, Madrid, Spain), anti-ACAT2, anti-MTP, and LXRα/β (Santa Cruz Biotechnology, Quimigen, Madrid, Spain). Membranes were incubated with peroxide-conjugated secondary antibodies for 1 h at room temperature. The chemiluminescence signal was detected using the ECL kit Select-kit (GE Healthcare, Madrid, Spain) and read in an ImageQuant LAS 500 (GE Healthcare, Madrid, Spain). Ponceau S staining (Sigma-Aldrich, Madrid, Spain) was used as the loading controls. The quantification of the protein levels was made using ImageQuant 5.0 software (GE Healthcare Life Sciences, Madrid, Spain) and expressed as a densitometry unit with respect to loading control.

### 2.9. Statistical Analysis

Statistical analysis was performed using SPSS version 28.0 (SPSS Inc., Chicago, IL, USA). The results were expressed as mean values ± SD. Continuous variables were compared using one-way ANOVA, followed by the Scheffe or Tamhane T2 post-hoc methods, depending on the assumption of equality or inequality of variances, respectively. For comparisons of non-parametric variables, the Kruskal–Wallis test was used followed by the Dunn–Bonferroni approach. Pearson product–moment correlations between parameters and Spearman correlations between scores and parameters were also determined. Statistical significance was considered *p* < 0.05. Graphs were drawn using GraphPad Prism version 9 (GraphPad software, Inc., La Jolla, CA, USA).

## 3. Results

### 3.1. Feed Intake, Ponderal Parameters, Total Feces, and Fat Excretion

Non-significant differences in daily feed intake, body weight gain, and small intestine weight were found (*p* > 0.05) (Table 1). However, significant differences were observed in cholesterol intake, liver weight, and total fecal and fat excretion (*p* < 0.001). Cholesterol intake was higher in the LD and LD-Si groups compared to the ED group (*p* < 0.0001). Liver weight was higher in the LD and LD-Si groups compared to the ED group (*p* < 0.001). Although there was a trend toward lower liver weight, no significant differences between LD and LD-Si rats were found (*p* > 0.05). In LD rats, total fecal excretion (40.6%) and fecal fat excretion (113.3%) were significantly higher (*p* < 0.05) compared to ED rats. The LD-Si group showed significantly higher fecal content (22.5%) and fecal fat excretion (48.9%) (*p* < 0.01) than the LD group. LD-Si rats showed the greatest fecal excretion, with a rate of 72.0%, and had the highest fecal fat content at 217.5%, significantly surpassing the ED group (*p* < 0.01).

### 3.2. Plasma Concentrations of Glucose, Insulin, HOMA-IR, Total Cholesterol, Triglycerides, VLDLtg, VLDLc, LDLc, IDLc, HDLc, and Atherogenic Index

Glycemia, insulinemia, HOMA-IR, triglycerides (TG), VLDLtg, total and lipoprotein cholesterol, and AI are shown in Table 2. When comparing LD to ED, LD rats showed significantly higher glycemia (*p* < 0.001), but lower levels of insulinemia and HOMA-IR (*p* < 0.001). LD-Si rats exhibited intermediate values, with significant differences compared to LD and ED rats (*p* < 0.001) regarding insulinemia, and only to LD in glycemia (*p* < 0.001). ED rats displayed higher TG levels than LD ones (*p* < 0.001). LD-Si showed lower TG plasma concentrations than the previous groups, being significantly different to them (*p* < 0.001). The LD group showed hypercholesterolemia (87.5% of LD rats), higher plasma VLDLc, IDLc, and LDLc, and lower VLDLtg and HDLc levels and as consequence higher AI compared to the ED group (*p* < 0.001). The elevated cholesterol levels in the LD groups were reversed after Si-RM consumption, reaching the values of the ED group. Silicon did not raise VLDLtg or HDLc levels, which remained at the same level as those in the LD group (*p* > 0.05), and lower than the ED group (*p* < 0.001). AI decreased in the LD-Si group, being significantly different in respect to both ED and LD groups (*p* < 0.001). In the LD-Si group, VLDLc showed similar levels to those of the ED group and significantly lower (*p* < 0.001) than those of the LD group. In the case of IDLc and LDLc, ED rats showed lower and significantly different values to both LD and LD-Si ones (*p* < 0.001).

### 3.3. Molecular Markers of Jejunal Cholesterol Absorption

Jejunal cholesterol absorption transporters are showed in Figure 2. The immunolocalization of NPC1L1, ACAT2 and MTP were measured by immunohistochemistry (Figure 2A,B). ACAT2 and MTP levels (Figure 2C,D, respectively) were also assayed by western blot. In all groups, the absorption cholesterol transporters’ immunoreactivities were located in the superficial epithelium along the villi and crypts, showing intensities from moderate to strong (Figure 2A,B). The LD group displayed higher NPC1L1, ACAT2, and MTP transporter levels than the ED group (*p* < 0.001). However, in LD-Si rats, although the NCP1L1 immunoreactivity remained high and at the same level as the LD group, ACAT2 (IRS = −21%, *p* < 0.0001; WB = −27%, *p* < 0.0001) and MTP (IRS = −22.5%; *p* < 0.001; WB = −27.5%, *p* < 0.0001) levels were significantly lower than those of LD and did not differ from ED (*p* < 0.05).

### 3.4. Cholesterol Efflux, Transintestinal Cholesterol Excretion (TICE), and Jejunal ABCA1 Levels

Figure 3A,B shows the immunohistochemical detection of ABCG5, ABCG8, ABCA1, LXRα/β, and LDLr proteins. ABCG5, ABCG8, and LXRα/β levels were also assayed by western blot (Figure 3C–E). Significant lower immunoreactivity scores in all proteins in LD group regarding ED group (*p* < 0.05) were observed. Despite LD-Si group presented lower ABCG5 (IRS = −12.6%, *p* = 0.033, WB = −21.1%, *p* = 0.005), ABCG8 (IRS = −20.7%, *p* = 0.02, WB = −24.2%, *p* = 0.0007) and ABCA1 (IRS = −32.66%, *p* < 0.0001) transporters’ levels respect to the ED group, Si-RM consumption induced increments of ABCG5 (IRS = 26.2%, *p* = 0.009, WB = 25.1%, *p* = 0.001), ABCG8 (31.4%, *p* < 0.001, WB = 43%, *p* = 0.02) and ABCA1 (IRS = 13.91%, *p* = 0.018) respect to the LD group. Likewise, the LD-Si group showed higher LXRα/β levels than LD group (IRS = 53.5%, *p* < 0.001, WB = 49.8%, *p* = 0.003), achieving similar values than ED (*p* > 0.05). ED group showed the highest levels of LDLr, being significantly different to LD (IRS = 38.5%; *p* = 0.003) and LD-Si (IRS = 24.24%, *p* = 0.032). LDLr levels in LD-Si were significant higher compared to those found in LD rats (*p* < 0.01).

### 3.5. Hepatic MTP, ABCA1, and HMGCR Levels

Figure 4A shows photographs representing the immunostaining of hepatic MTP, ABCA1, and HMGCR proteins. ED and LD rats showed an intense staining of MTP protein level in the cytoplasm of the hepatocytes distributed across the surface (Figure 4A). The LD group showed the highest MTP scores (LD vs. ED IRS = 12.4%, *p* = 0.014) (Figure 4B). In the livers of LD-Si rats, there was a weak MTP staining especially in centrilobular hepatocytes, and it obtained the lowest MTP levels compared to the ED (IRS = −24.5%, *p* < 0.0001) and LD groups (IRS = −33.9%, *p* < 0.0001). The ABCA1 immunoreactivity was lower in the LD group compared with the ED group (IRS = −44.85%, *p* < 0.0001). The LD-Si group showed similar ABCA1 levels to LD (*p* > 0.05) (Figure 4C). HMGCR immunoreactivity scores were higher in the LD and LD-Si groups compared with the ED group (IRS = 45%, *p* < 0.001). Non-significant differences were found between LD and LD-Si (*p* > 0.05) (Figure 4D).

### 3.6. Morphometric Parameters of the Absorptive Area, Epithelium Turnover, Differentiation, and Barrier Integrity of the Jejunum

Morphologic parameters of the jejunal mucosa are shown in Table 3. The LD group showed significantly higher villi width (14%, *p* = 0.023) and area (15%, *p* = 0.021) than those of the ED group. Silicon significantly decreased the villi width (−16.4%, *p* = 0.004) and the absorptive area (−22.8%, *p* = 0.001) compared to LD, achieving the ED values (ED vs. LD-Si, *p* > 0.05). Conversely, in crypt depth, LD-Si rats showed deeper crypts than ED rats (19.5%, *p* < 0.0001). The villi height and the ratios of villi height/width and villi height/crypt depth did not show differences between the groups (*p* > 0.05). No significant differences in the number of villi and crypts and PAS and AB goblet cells among experimental groups were observed (*p* > 0.05).

Figure 5 shows the immunohistochemistry photographs (Figure 5A) and the values of the intestinal proliferation and apoptosis measured by PCNA-LI (Figure 5B) and TUNEL index (Figure 5C). The LD group showed an increased epithelial turnover in the jejunal mucosa, with a higher PCNA-LI in the crypts (26.3%, *p* = 0.021) and a greater number of positive TUNEL cells in the villi (57.1%, *p* = 0.0006) respect to ED group. Compared to the ED and LD groups, the LD-Si group increased the mitotic activity in the crypts, with higher levels in both the PCNA-LI (ED vs. LD-Si; 35.3%, *p* = 0.0037, and LD vs. LS-Si; 12.1%, *p* = 0.034), and the number of positive TUNEL nuclei (ED vs. LD-Si; 67.7%, *p* < 0.0001, and LD vs. LD-Si; 24.7%, *p* = 0.024), favoring the renewal of the absorptive surface.

Figure 5D–F shows photographs and the levels of Occludin and Claudin 1 immunoreactivity scores of the mucosa of the jejunum, respectively. A differential pattern on the immunoreactivity of the tight junctions’ protein levels was observed. The ED group showed higher values in Occludin (*p* = 0.002) and Claudin 1 (*p* = 0.046) than the LD group, indicating an altered jejunal absorptive barrier in late-stage T2DM rats. Although the LD-Si group did not significantly differ from Claudin 1 respect to LD, Occludin levels were significantly higher than those of LD, achieving ED values (*p* > 0.05). Furthermore, LD-Si rats maintained lower Claudin 1 levels compared to those of the ED group (*p* = 0.01).

### 3.7. Heatmap

Figure 6 summarizes all significant Pearson or Spearman correlations found between the different markers involved in cholesterol absorption and metabolism, T2DM and lipid profile markers, as well as fecal fat content. Significant differences are highlighted and marked with asterisks.

NCP1L1 protein levels appears positively associated with total cholesterol (r = 0.690; *p* = 0.004), LDLc (r = 0.832; *p* = 0.0001) and fecal fat (r = 0.70, *p* = 0.004) but negatively with HDLc (r = −0.532: *p* = 0.044), TG (r = −0.535: *p* = 0.04) and VLDLtg (r = −0.579, *p* = 0.024). In addition, jejunal ACAT2 and MTP proteins were associated positively with total cholesterol (ACAT2: r = 0.828, *p* = 0.0001; MTP: r = 0.670, *p* = 0.002), VLDLc (ACAT2: r = 0.829, *p* = 0.0001; MTP: r = 0.839, *p* = 0.002), and LDLc (ACAT2: r = 0.891, *p* = 0.0009; MTP: r = 0.549, *p* = 0.018), but HDLc, TG, and VLDLtg levels were not associated with ACAT2 and MTP proteins (*p* > 0.05). Regarding cholesterol efflux, the insulinemia, HOMA-IR, total cholesterol, VLDLc, and LDLc levels were inversely correlated to both jejunal ABCG5/G8 and LDLr levels (*p* < 0.03 to *p* < 0.0001). Non-significant correlations between HDLc, TG, and VLDLtg levels and ABCG5/8, LXRα/β, and LDLr levels (*p* > 0.05) were found. In addition, ABCG5/G8 levels were stimulated by LXRα/β as suggested by the significant and positive correlation coefficients (ABCG5: r = 0.744, *p* = 0.0002; ABCG8: r = 0.577, *p* = 0.013). The ABCG5/8 levels were also correlated to those of LDLr, confirming its implication in the TICE (ABCG5: r = 0.883, *p* = 0.0001; ABCG8: r = 0.789, *p* = 0.001). To verify the involvement of ABCA1 in the RCT, the correlations between both jejunal and liver ABCA1 and the plasma cholesterol and lipoproteins levels were studied. ABCA1 protein correlated positively with circulating HDLc levels (jejunum: r = 0.639, *p* = 0.002; liver: r = 0.607, *p* = 0.002). Total cholesterol, VLDLc, LDLc levels were inversely correlated to both jejunal and liver ABCA1 and jejunal LDLr levels (*p* < 0.002 to *p* < 0.0001). In addition, in the jejunum, LXRα/β was a significant contributor to ABCA1 levels as shown by the significant and positive correlation coefficients (r = 0.645, *p* = 0.005). Fecal fat content was inversely correlated with hepatic and jejunal ABCA1, jejunal LDLr, hepatic MTP, TG, VLDLc, HDLc, and HOMA-IR and positively correlated with AI, LDLc, and IDLc (*p* < 0.01 to *p* < 0.0001). The involvement of hepatic MTP in the formation of VLDL was confirmed by the positive correlation between both parameters (r = 0.766, *p* < 0.0001). In addition, a positive significant correlation between jejunal MTP level and plasma cholesterol was found (r = 0.496, *p* = 0.04). Diabetic dyslipidemia was confirmed by the significant correlations between insulinemia, HOMA-IR, and glycemia with total cholesterol, TG, VLDLc, LDLc, and AI (*p* < 0.01 to *p* < 0.0001).

## 4. Discussion

Recently, attention has been focused on the search for new dietary ingredients with cholesterol-lowering potential capable of being used in the treatment of diabetic dyslipidemia. There are relatively few studies that have identified the beneficial effect of silicon against hyperglycemia and hyperlipidemia, suggesting that it could be used as a useful natural product in the treatment of T2DM. In the present study we focus on investigating the Si-RM effects on metabolic processes involved in net cholesterol efflux in the jejunum and liver. The main results confirmed that Si-RM consumption, produced: (a) lower levels of intestinal ACAT2 and MTP enzymes which could avoid the overproduction of jejunal chylomicrons (CM) partially blocking the cholesterol absorption; (b) a minor villous area of absorption that maintained jejunal barrier integrity and epithelial villi regeneration; (c) jejunal upregulation of LXRα/β and ABCG5/8 levels facilitating the output of the excess dietary cholesterol absorbed from the intestinal lumen; (d) higher ABCA1 levels allowing the transference of the excess cholesterol from the intestine to HDL particles; (e) higher jejunal LDLr levels activating endogenous cholesterol efflux by TICE; and (f) a promotion of the hepatic MTP downregulation decreasing the formation of atherogenic cholesterol-enriched VLDL particles.

As we have previously shown [25,26], progression to a late-stage model of T2DM was successfully established in LD rats, being a suitable animal model for studies on the impact of Si-RM intake on pathophysiological changes of diabetic dyslipidemia. This was evidenced by marked hyperglycemia and significantly lower insulin production and HOMA-IR, due to both the STZ inducing atrophy of the pancreatic islets [26,31] and excess dietary cholesterol inhibiting insulin secretion from β cells [11,32]. Also, our dyslipemic T2DM animal model (LD group) was evidenced by higher total cholesterol, VLDLc, IDLc, LDLc levels, lower HDLc, and therefore higher AI levels, compared to an early-stage T2DM (ED group). Reduced plasma TG in metabolic syndrome rats fed a HSFD has previously been described [33]. Low plasma TG levels found in LD rats might reflect an HSFHCD-induced partitioning of lipids into the liver, either by reduced hepatic TG production or by increased TG clearance [34] and/or by the higher beta-VLDL particle synthesis in HSFHCD-fed animals [23,25,29].

In the present study, the higher total cholesterol and atherogenic lipoprotein levels in LD rats vs. ED rats seems to result from an increase in intestinal absorption of dietary cholesterol caused by higher NPC1L1, ACAT2, and MTP protein levels. The high expression of NPC1L1 found in LD rats reflects the higher cholesterol uptake in enterocytes. NPC1L1 upregulation favors the prime import of cholesterol into enterocytes, while ACAT2 and MTP enzymes esterify free cholesterol (CE) for packing into CM and transfer them into the lymphatics [35]. In T2DM, CM production is a major player in the atherogenic process, and it is increased through disturbance in cholesterol absorption [36]. Altered upregulation of intestinal transporters, in particular NPC1L1 and ACAT2 and MTP enzymes, has been shown in both animal and human T2DM [11,36], just as it is found in the LD group. On the contrary, these results differ from those found previously by our research group in the duodenum since no significant differences in the cholesterol absorption markers were observed between the ED and LD groups [26]. The resulting differences could be attributed to the fact that while lipid digestion truly starts in the duodenum, and depends on an intricate interplay between pancreatic lipase, colipase, and bile salts, the maximum lipid absorption (95%) occurs in the proximal jejunum [37]. In just a few days the jejunum can adapt its lipid absorption capability in response to the intake of an HSFD (60%) [38] and/or high glucose concentrations [39] by differentially expressing the largest number of genes [37]. In addition, LD rats showed a higher absorption area with a greater width of the villi leading to enhanced lipid uptake by lymphatic vessels, which may improve nutrient absorption. Moreover, an altered intestinal barrier by lower Occludin (76%) and Claudin1 (20%) expression, greater cell proliferation (40%), and apoptosis (137%) together with an accelerated cell turnover had been shown in the jejunum of LD rats. Alterations in the cell proliferation rate [40], cell number, villi length and thickness, and crypt depth have been observed in T2DM rats [41,42,43], likely as efforts to expand the intestinal absorptive surface area. Altogether, this could contribute to the increased lipid absorption capability observed and, therefore, the hypercholesterolemic state observed in late-stage T2DM rats. This fact suggests that intestinal cholesterol absorption inhibitors could be effective in lowering lipids and reducing the risk of CV diseases [44].

In addition, LD rats showed downregulation of LXRα/β with respect to ED which in turn reduced levels of ABCG5/8 proteins. LXRα/β-ABCG5/G8 pathway is one of the strategies to regulate intestinal cholesterol homeostasis [45]. An enhancement of CM secretion in individuals with T2DM has been related to low ABCG5 and ABCG8 mRNA levels [46,47]. Despite the higher NCLPL1 and lower ABCG5/8 levels found, LD rats showed an increased steatorrhea compared to ED rats. This fact could be related to an excess of ingested cholesterol (1.2%) in the LD group, since Petit et al. [48] found a higher expression of proteins involved in cholesterol uptake by the jejunal mucosa, without steatorrhea, in mice fed a high-fat diet without added cholesterol. Furthermore, it could happen that the amount of fat in the diet exceeds the absorptive capacity of the intestine and, therefore, saturates the mechanism, increasing the amount of fat excreted in the feces. In accordance with this, the amount of fecal fat was not significantly correlated with ABCG5/G8 but appears related with NPC1L1 (*p* < 0.01) confirming that the amount of fecal fat does not appear to be directly regulated by intestinal efflux transporters. In parallel, LD rats showed lower jejunal LDLr levels compared to ED rats, which in turn could decrease TICE. Furthermore, we have found significant negative correlations between both LDLr and ABCG5/8 levels with total cholesterol, VLDLc, IDLc, and LDLc levels in line with the suggestion that TICE relies primarily on the uptake of atherogenic apoB-containing lipoproteins by LDLr into the basolateral membrane of enterocytes and luminal excretion via apical transporters ABCG5/G8 [49]. This allows total cholesterol to be transported through the blood to the intestinal cells and eventually be secreted into the intestinal lumen, contributing ~35% to cholesterol excretion in the feces [50,51,52]. Loss of LDLr function and reduction of intestinal ABCG5/G8 levels could contribute to an increase in cholesterolemia in LD rats by decreasing both luminal dietary cholesterol excretion and cholesterol reabsorption through TICE [53]. Regarding this, it has been reported that mice fed a Western diet and HSFD increased TICE by 1.5. times and two times, respectively, while a high-cholesterol diet had no effect on activating TICE, favoring an atherogenic lipidic profile [54].

Furthermore, the excess cholesterol removal by HDLc was compromised in LD rats. Jejunal and hepatic ABCA1 levels were lower in LD rats vs. ED rats as a consequence of LXRα/β downregulation [45]. In addition, significantly lower HDLc levels were found in LD rats (16%, *p* < 0.01). ABCA1 expression is known to strongly influence the level of plasma HDLc [55], supporting our results. Moreover, positive relationships between both jejunal and liver ABCA1 with circulating HDLc levels and negative correlations with higher IDLc and LDLc levels have been found in LD rats. This could result from a lower efflux of intracellular free cholesterol and phospholipids across the hepatic and intestinal plasmatic membrane to be combined with apolipoprotein A-I (ApoA-I), to form nascent HDL particles followed by blocking the first step of reverse cholesterol transport (RCT) [55,56]. The liver selectively absorbs lipids from HDL particles through SR-B1 and transfers CE to bile for excretion through the intestines, thereby completing the entire RCT process [57]. According to our previous results [25], similar hepatic SR-B1 levels found in both rat groups could explain an abnormal RCT pathway altering cholesterol metabolism [58]. Therefore, the levels of HDLc were not balanced to bear the pressure of RCT in LD rats. Decreased ABCA1 expression in the cholesterol efflux impairment has been reported to be closely associated with T2DM and CV risk factors [59,60].

In addition to intestinal absorption and biliary and fecal excretion, hepatic de novo synthesis is mainly involved in cholesterol homeostasis. HMGCR is the rate-limiting enzyme involved in cholesterol synthesis and contributes to 60–70% of the total cholesterol pool [11,61]. In our study, HMGCR was 43% higher in LD rat liver compared to ED counterparts. HMGCR overexpression was directly correlated with LDLc and inversely with HDLc in LD rats. As a direct effect of increased cholesterol synthesis, the secretion of VLDLc and the conversion to LDLc was increased. Human studies have shown that cholesterol synthesis was inevitably increased in a high-cholesterol environment in diabetic patients [62,63].

Several studies have detected that liver MTP expression is upregulated by the dietary fat/cholesterol content, T2DM, and insulin resistance states, as well as disruption in insulin signaling pathways leading to an overproduction of VLDL particles [64,65,66]. We also found significant increases of 16% in hepatic MTP levels of LD rats compared to ED rats, which were parallel with decreases in TG and VLDLtg, although they were not significantly correlated.

A significant correlation between plasma insulin and HOMA-IR with plasma cholesterol, TG, VLDLc, IDLc, LDLc, and AI have been found. In parallel, PI3K and pAktSer473 levels were negatively correlated with VLDLc [25], confirming that an altered InsRbeta/PI3K/AKT insulin signaling pathway could be implicated in the dyslipidemia observed in LD rats. The above findings, associated with the preponderance of dense small and reduced large cholesterol-rich HDL, are predominant features of diabetic dyslipidemia in LD rats [67].

Si-RM consumption had an anti-diabetogenic effect leading to less progression of T2DM by improving the atherogenic lipid profile observed in LD rat counterparts. One of the mechanisms involved in the cholesterol-lowering effect of silicon was associated with the inhibition of the intestinal cholesterol absorption. Our findings align with previous research conducted on the duodenum, where Si-RM favorably inhibited the jejunal levels of ACAT2 and MTP enzymes, but not of NPC1L1, accompanied by the reduction of plasma cholesterol, VLDLc, IDLc, and LDLc levels compared to LD rats. The decrease of intestinal ACAT2 and MTP levels could significantly reduce CE and its transport within CM. Selective inhibition for intestinal ACAT-2 and MTP has been demonstrated to ameliorate diabetes-induced dyslipidemia in humans [68,69,70,71]. In addition, the morphometric changes induced by Si-RM were more evident in the jejunum than in the duodenum, as a consequence of the fact that maximum lipid absorption (95%) occurs in the proximal jejunum [37]. LD-Si rats showed significantly lower absorptive areas and higher cell renewal in the absorptive epithelium vs. the LD group, comparable to those of the ED group. These results suggest that the LD-Si group exerted an adaptive jejunal response with lower villi width and area, but higher epithelial turnover and crypt depth, promoting continuous intestinal epithelial renewal through multiplication and differentiation of the villi stem cells at the basal area of the crypt, and the migration of mature cells along the villus. The significant increase in Occludin levels observed in LD-Si rats could contribute to improving the intestinal barrier integrity and to reducing increased intestinal permeability, preventing possible T2DM chronic low-grade inflammation [43]. High cell turnover and epithelial integrity should contribute to the adaptive capacity of the LD-Si group to reduce the T2DM-induced intestinal absorptive barrier observed in LD rats. The above results confirm that Si-RM significantly reduced cholesterol absorption capacity and could effectively block hypercholesterolemia, slowing down progression to late-stage T2DM.

In addition to the downregulation of jejunal MTP, LD-Si rats also showed lower hepatic MTP expression compared to LD rats, leading to a decrease in the VLDLc secretion rate [72]. Positive significant relationships between both jejunal and hepatic MTP with VLDLc levels have been found. Although the inhibition of MTP in enterocytes may contribute to reducing plasma cholesterol levels by blocking CM formation, the hepatic MTP inhibition also contributes by reversing the production and secretion of VLDL in LD rats. Because insulin downregulates MTP [73], we hypothesized that the effect of Si-RM on decreasing hepatic VLDL production could be due to a direct effect by blocking MTP [72] or indirectly favoring pancreatic insulin secretion as indicated by increased HOMA-β [26] and a positive regulation of the insulin signaling pathway [25]. Unfortunately, in humans, MTP inhibitor drugs cause severe hepatic lipid accumulation due to their pharmacological effect inhibiting hepatic VLDL secretion, and it still represents a major safety concern [66]. In fact, Si-RM intake did not reduce the liver weight of the LD-Si group compared to the LD group. Studies are necessary to evaluate the degree of liver damage that could be produced by the accumulation of fat in the liver, as a consequence of MTP inhibition.

Silicon-enriched meat effects also targeted cholesterol efflux transporters. Si-RM consumption increased jejunal LXRα/β expression and, consequently, ABCG5/G8 levels were higher, reaching those of the ED group. Just as happened in the duodenum [26], Si-RM allowed an increase in the luminal dietary cholesterol excretion. These results correlated inversely with plasma glucose, cholesterol, VLDLc, IDLc, LDLc, and AI levels, indicating its contribution to improving the diabetic dyslipemia observed in LD rats. In parallel, the upregulation of jejunal LXRα/β by Si-RM consequently increased LDLr levels by 23% compared to the LD group, achieving the levels of ED group. LDLr correlated positively with ABCG5/G8 and insulinemia, but inversely with plasma glucose, cholesterol, VLDLc, IDLc, LDLc, and AI. These findings suggest that Si-RM activated the transporters responsible for TICE in a compensatory mechanism of cholesterol excretion, thus contributing, at least partially, to the clearance of atherogenic particles and to the increased fecal fat excretion observed in LD-Si rats. In this regard, fat presence in feces was 1.5 and 3-fold higher in LD-Si than LD and ED groups, respectively [26]. However, no significant relationship was found between jejunal LDLr levels and the amount of fat in feces. Therefore, other hypotheses should not be disregarded [26].

However, Si-RM partially affected cholesterol efflux mediated by ABCA1, since significantly higher ABCA1 levels in LD-Si compared to LD rats (17%, *p* < 0.01) were only found in the jejunum, without changes in the liver. It is clear, therefore, that if the expression of ABCA1 can be upregulated by Si-RM, HDLc levels should increase, having a favorable health effect reversing the atherosclerotic lipoprotein profile [60,74]. However, in our study higher intestinal ABCA1 levels were not enough to enhance plasma HDLc levels in LD-Si rats, which were similar to those in LD rats (*p* > 0.05) and lower than those in ED rats (*p* < 0.01). Consequently, RCT promoted by HDLc could be compromised in LD-Si rats, as indicated by the similar hepatic SR-B1 levels in all experimental groups found in our previous studies [25]. Silicon-induced ABCA1 activation in the jejunum could be due to a negative feedback mechanism induced by the upregulation of LXRα/β and/or low plasma HDLc levels. Nonetheless, the precise mechanism involved in the silicon-induced ABCA1 upregulation needs to be clarified in future research.

Finally, no changes in HMGCR liver levels were found between the LD and LD-Si groups, indicating that Si-RM’s cholesterol-lowering effect was not associated with cholesterol biosynthesis. Statins reduce plasma cholesterol by inhibiting HMG-CoAR [75,76], although they increase the risk of T2DM in a dose-dependent manner [77]. This limits the use of statins and suggests the need for alternative and/or complimentary therapeutic approaches in diabetic patients [78]. In this sense, co-administration of Si-RM intake and low doses of statins could have the potential benefit of attenuating statin-induced diabetogenic side effects in advanced T2DM patients.

In short, the dietary cholesterol absorption into intestinal epithelial cells was inhibited by Si-RM, while excretion was increased, facilitating cholesterol efflux, TICE, and fecal fat excretion. We have demonstrated that Si-RM mainly exerts the hypocholesterolemic effect in two ways, which included firstly the limitation of excessive cholesterol absorption by inhibition the intracellular cholesterol into CE in an inactivation way of jejunal ACAT2 and blocked MTP CM secretion with a decrease in intestinal absorptive area, and secondly Si-RM upregulated LXRα/β which in turn induced higher LDLr and ABCG5/G8 levels, diverting more dietary cholesterol from the enterocyte back into the intestinal lumen and activating TICE. In addition, Si-RM has turned out to be a potent MTP inhibitor with the potential to block both hepatic and intestinal apoB-containing lipoprotein assembly and consequently lower plasma TG, cholesterol total, IDLc, LDLc, and VLDLc.

## 5. Conclusions

This study has shown relevant results and has exposed for the first time the mechanisms by which silicon consumption, as a functional meat food ingredient, mitigated dyslipidemia associated with late-stage T2DM by improving the impaired of cholesterol metabolism. These results suggest that Si-RM consumption may be considered as a nutritional therapeutic strategy in the management of diabetic dyslipidemia. Silicon could be used alone or in conjunction with other cholesterol-lowering pharmacologic treatments to improve T2DM and associated diseases. Future research may be essential to substantiate these findings and explore dietary alternatives, making meat a healthier product with potential therapeutic benefits and also evaluating the safety profiles of long-term silicon consumption.

## Figures and Tables

**Figure 1 foods-13-01794-f001:**
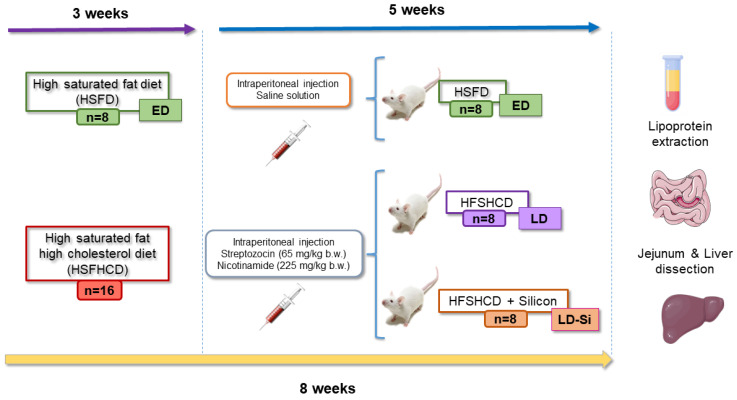
Experimental design: The ED group consisted of early-stage T2DM control rats fed a high-saturated-fat diet (HSFD). LD group included late-stage T2DM rats fed a high-saturated-fat, high-cholesterol meat-based diet and given injections of streptozotocin and nicotinamide (STZ/NAD). The LD-Si group comprised late-stage T2DM rats fed a high-saturated-fat, high-cholesterol diet enriched with silicon and given injections of STZ/NAD. Each group contained eight rats (n = 8/group).

**Figure 2 foods-13-01794-f002:**
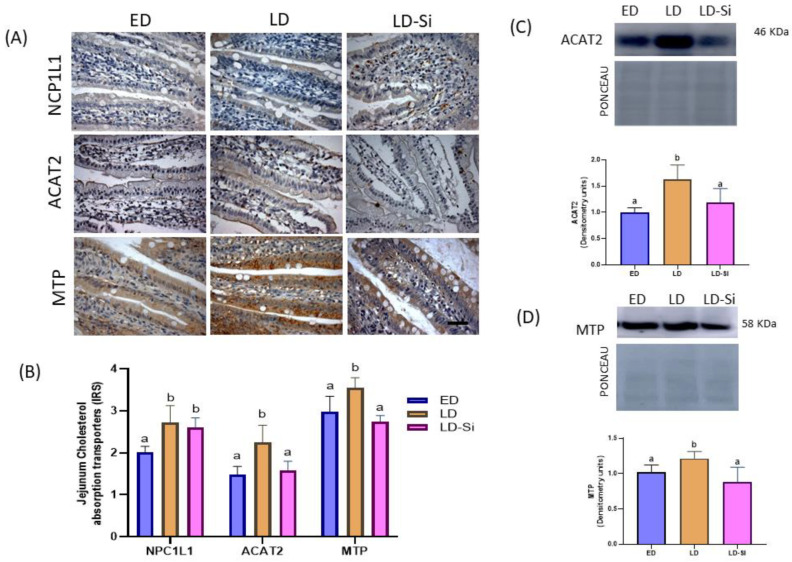
Effect of the silicon-enriched meat (Si-RM) on NPC1L1, Niemann-Pick C1-Like 1; ACAT2, acetyl-Coenzyme A acetyltransferase-2; MTP, microsomal triglyceride transfer protein of the jejunal epithelium in the early-stage diabetes (ED), late-stage diabetes (LD), and late-stage diabetes-silicon (LD-Si) groups. (**A**) Representative images of immunohistochemistry labelling of NPC1L1, ACAT2 and MTP. Scale-bar: 50 µm. (**B**) Immunoreactivity scores (IRS) of NPC1L1, ACAT2 and MTP. (**C**) Bands of western blot and percentage data of densitometric quantification for jejunal ACAT2. (**D**) Bands of western blot and percentage data of densitometric quantification for jejunal MTP. Values expressed as mean ± SD (IHC n = 8/group; WB n = 4/group). Different letters (a < b) indicate significant differences between groups, ANOVA (*p* < 0.05 followed by Scheffe or Tamhane post-hoc test).

**Figure 3 foods-13-01794-f003:**
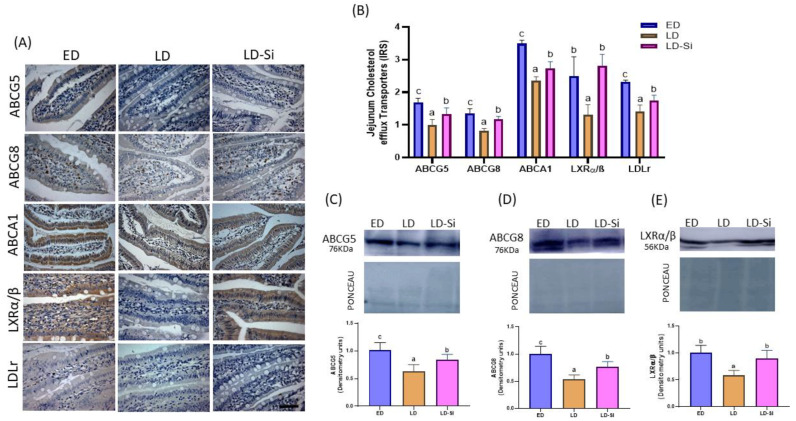
Effect of the silicon-enriched meat (Si-RM) on ATP-binding cassette subfamily G members 5 (ABCG5) and 8 (ABCG8) and subfamily A1 (ABCA1) transporters, the liver X receptor transcription factor (LXRα/β) and low-density lipoprotein receptor (LDLr) of the jejunal epithelium in the early-stage diabetes (ED), late-stage diabetes (LD), and late-stage diabetes-silicon (LD-Si) groups. (**A**) Representative images of immunohistochemistry labelling (Scale-bar: 50 µm), (**B**) immunoreactivity scores of ABCG5, ABCG8, ABCA1, LXRα/β and LDLr, (**C**) Bands of western blot and percentage data of densitometric quantification for jejunal ABCG5, (**D**) Bands of western blot and percentage data of densitometric quantification for jejunal ABCG8, (**E**) Bands of western blot and percentage data of densitometric quantification for jejunal LXRα/β.Values expressed as mean ± SD (IHC n = 8/group; WB n = 4/group). Different letters (a < b < c) indicate significant differences between groups, ANOVA (*p* < 0.05 followed by Scheffe or Tamhane post-hoc test).

**Figure 4 foods-13-01794-f004:**
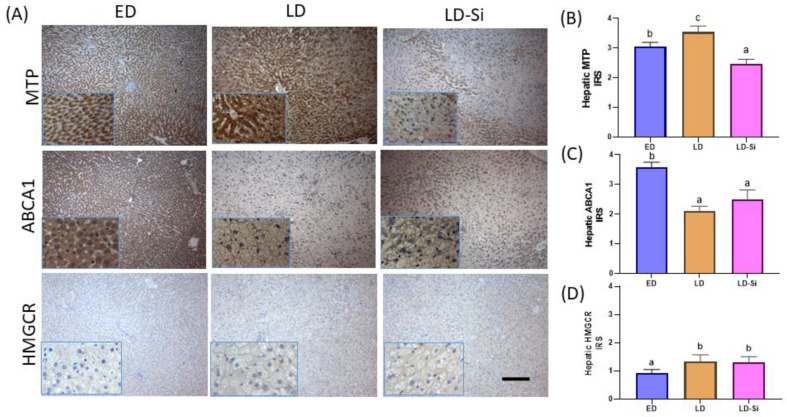
Effect of the silicon-enriched meat (Si-RM) on microsomal triglyceride transfer protein (MTP), ATP-binding cassette subfamily A member 1 transporters (ABCA1) and 3-hydroxy-3-methylglutaryl CoA reductase (HMGCR) of the liver in the early-stage diabetes (ED), late-stage diabetes (LD), and late-stage diabetes-silicon (LD-Si) groups. (**A**) Representative images of immunohistochemistry labelling of MTP, ABCA1 and HMGCR. Scale-bar: 50 µm, insert: 20 µm. Immunoreactivity scores of (**B**) MTP, (**C**) ABCA1, and (**D**) HMGCR. Values expressed as mean ± SD (n = 8/group). Different letters (a < b < c) indicate significant differences between groups, ANOVA (*p* < 0.05 followed by Scheffe or Tamhane post-hoc test).

**Figure 5 foods-13-01794-f005:**
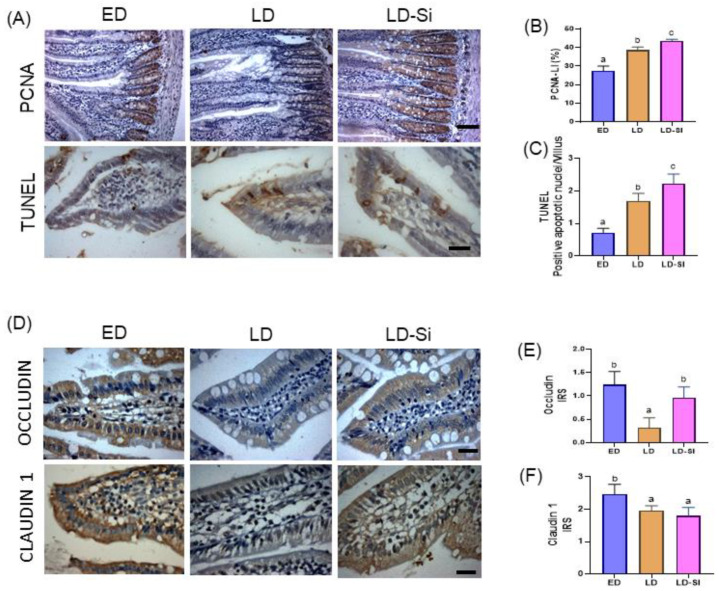
Modifications in cell proliferation, apoptosis, and barrier integrity of the jejunal epithelium were examined in early-stage diabetes (ED), late-stage diabetes (LD), and late-stage diabetes-silicon (LD-Si) groups. (**A**) Representative images show immunohistochemical staining for proliferating cell nuclear antigen (PCNA) (scale bar: 100 µm) and terminal deoxynucleotidyl transferase dUTP nick end labeling (TUNEL) (scale bar: 10 µm). Immunoreactivity scores for (**B**) PCNA-LI and (**C**) TUNEL. (**D**) Representative images show immunohistochemical staining for Occludin and Claudin 1 (scale bar: 50 µm). Immunoreactivity scores for (**E**) Occludin and (**F**) Claudin 1. Values are expressed as mean ± SD (n = 8/group). Different letters (a < b < c) indicate significant differences between groups (ANOVA, *p* < 0.05, followed by Scheffe or Tamhane post-hoc test).

**Figure 6 foods-13-01794-f006:**
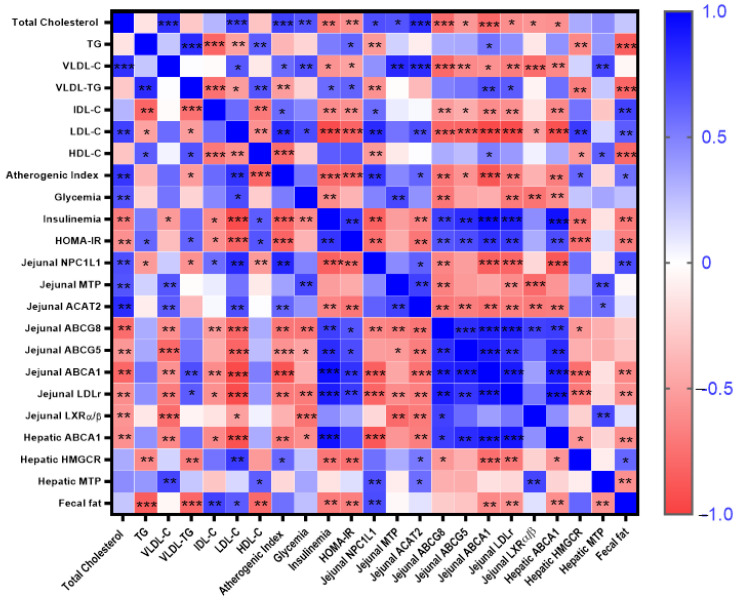
Interactions between glycemic and dyslipemic status, fecal fat content, and jejunal and hepatic cholesterol transporters and enzymes. Pearson and Spearman correlation values were used for the matrix. The parameters displayed in the heatmap are total cholesterol, TG, VLDLc, VLDLtg, IDLc, LDLc, and HDLc, atherogenic index (AI), glycemia, insulinemia, HOMA-IR; jejunal NPC1L1, ACAT2, MTP, ABCG8, ABCG5, ABCA1, LDLr, LXRα/β, and hepatic ABCA1, HMGCR, MTP, and fecal fat content. The color intensity of the heatmap represents the association degree: Blue, positive and Red, negative. * *p* < 0.05, ** *p* < 0.01, *** *p* < 0.001.

**Table 1 foods-13-01794-t001:** Daily total and cholesterol intake, body weight gain, small intestine and liver weight parameters, and total and fat fecal excretion of early-stage diabetes (ED), late-stage diabetes (LD), and late-stage diabetes-silicon (LD-Si) groups.

	ED Group	LD Group	LD-Si Group	*p*
Daily total intake (g/day)	17.8 ± 1.20	16.5 ± 0.55	17.1 ± 0.55	NS
Daily cholesterol intake (mg/day)	3.6 ± 0.36 ^a^	154.0 ± 12.1 ^b^	167.1 ± 8.5 ^b^	<0.0001
Body weight increase (g)	134.4 ± 25	117.3 ± 32.3	127.6 ± 34.6	NS
Small intestine weight (g)	1.86 ± 0.16	1.90 ± 0.27	1.86 ± 0.21	NS
Liver weight (g)	10.34 ± 0.82 ^a^	18.43 ± 2.38 ^b^	17.43 ± 2.42 ^b^	<0.001
Fecal excretion (g/day dry matter)	1.14 ± 0.08 ^a^	1.60 ± 0.06 ^b^	1.96 ± 0.24 ^c^	<0.0001
Fecal fat excretion (mg/g dry matter)	80.0 ± 6.7 ^a^	170.6 ± 10.1 ^b^	254.0 ± 14.3 ^c^	<0.0001

Values are presented as mean ± SD (n = 8/group). Different letters (a < b < c) denote significant differences between groups (*p* < 0.05, determined by ANOVA followed by Scheffe or Tamhane post-hoc test). NS indicates no significant differences between groups.

**Table 2 foods-13-01794-t002:** Plasma glucose, insulin, HOMA-IR, total and lipoprotein cholesterol concentrations, triglycerides and VLDLtg levels, and atherogenic index of early-stage diabetes (ED), late-stage diabetes (LD), and late-stage diabetes-silicon (LD-Si) groups.

	ED Group	LD Group	LD-Si Group	*p*
Glucose (mmol/L)	13.92 ± 0. 91 ^a^	18.11 ± 1.65 ^b^	15.26 ± 2.07 ^a^	<0.001
Insulin (μUI/mL)	15.84 ± 0.73 ^c^	5.41 ± 1.23 ^a^	8.11 ± 1.79 ^b^	<0.0001
HOMA-IR	9.79 ± 0.64 ^b^	4.57 ± 1.55 ^a^	5.87 ± 1.5 ^a^	<0.001
Cholesterol (mmol/L)	2.05 ± 0.07 ^a^	2.79 ± 0.05 ^c^	2.24 ± 0.04 ^b^	<0.0001
Triglycerides (mmol/L)	1.79 ± 0.25 ^c^	0.88 ± 0.11 ^b^	0.60 ± 0.16 ^a^	<0.001
VLDLc (mmol/L)	0.42 ± 0.05 ^a^	0.74 ± 0.09 ^b^	0.39 ± 0.07 ^a^	<0.001
VLDLtg (mmol/L)	1.57 ± 0.05 ^b^	0.31 ± 0.07 ^a^	0.21 ± 0.02 ^a^	<0.0001
LDLc (mmol/L)	0.03 ± 0.003 ^a^	0.33 ± 0.05 ^b^	0.23 ± 0.03 ^b^	<0.001
IDLc (mmol/L)	0.02 ± 0.003 ^a^	0.52 ± 0.03 ^b^	0.60 ± 0.11 ^b^	<0.001
HDLc (mmol/L)	1.45 ± 0.14 ^b^	1.07 ± 0.18 ^a^	1.10 ± 0.07 ^a^	<0.0001
AI (Non-HDLc/HDLc)	0.43 ± 0.22 ^a^	1.48 ± 0.45 ^c^	1.04 ± 0.15 ^b^	<0.001

Values are presented as mean ± SD (n = 8/group). Different letters (a < b < c) denote significant differences between groups (*p* < 0.05, determined by ANOVA followed by Scheffe or Tamhane post-hoc test). NS indicates no significant differences between groups. HOMA-IR, homeostatic model assessment of insulin resistance; VLDLc, very-low-density lipoprotein cholesterol; VLDLtg, very-low-density lipoprotein triglycerides; LDLc, low density lipoprotein cholesterol; IDLc, intermediate density lipoprotein cholesterol; HDLc, high density lipoprotein cholesterol; AI, atherogenic index (Non-HDLc/HDLc).

**Table 3 foods-13-01794-t003:** Morphometric parameters of jejunum of early-stage diabetes (ED), late-stage diabetes (LD), and late-stage diabetes-silicon (LD-Si) groups.

	ED Group	LD Group	LD-Si Group	*p*
Villi height (µm)	959.5 ± 113.9	1003.5 ± 101.1	920.0 ± 107.2	NS
Villi width (µm)	251.2 ± 16.8 ^a^	292.5 ± 24.4 ^b^	244.6 ± 25.4 ^a^	<0.01
Villi area (mm^2^)	0.79 ± 0.05 ^a^	0.85 ± 0.13 ^b^	0.71 ± 0.04 ^a^	<0.001
Crypt depth (µm)	284.4 ± 23.6 ^a^	315.7 ± 58.7 ^ab^	354.5 ± 24.5 ^b^	<0.01
Villi height/width	3.29 ± 0.30	3.16 ± 0.47	2.79 ± 0.24	NS
Villi height/crypt depth	3.69 ± 0.50	3.54 ± 0.23	3.38 ± 0.26	NS
Villi PAS goblet cells (no. positive cells)	27.18 ± 4.34	25.77 ± 6.57	25.75 ± 6.54	NS
Villi AB goblet cells (no. positive cells)	32.95 ± 5.35	32.19 ± 3.32	27.70 ± 3.47	NS
Crypt PAS goblet cells (no. positive cells)	13.95 ± 1.64	15.35 ± 1.58	15.70 ± 1.65	NS
Crypt AB goblet cells (no. positive cells)	14.33 ± 2.87	15.28 ± 2.19	14.45 ± 2.02	NS

Values are presented as mean ± SD (n = 8/group). Different letters (a < b) denote significant differences between groups (*p* < 0.05, determined by ANOVA followed by Scheffe or Tamhane post-hoc test). NS indicates no significant differences between groups. PAS, periodic acid Schiff; AB, Alcian Blue.

## Data Availability

The original contributions presented in the study are included in the article/Appendix A, further inquiries can be directed to the corresponding authors.

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
