# Peer review of "Silicon as a Functional Meat Ingredient Improves Jejunal and Hepatic Cholesterol Homeostasis in a Late-Stage Type 2 Diabetes Mellitus Rat Model"

_foods, 2024, doi:10.3390/foods13121794_

Round 1

Reviewer 1 Report

Comments and Suggestions for Authors

Dear Authors,

I have read and revised manuscript foods-3028811, titled: Silicon as a meat functional ingredient improves jejunal and hepatic cholesterol homeostasis in a late-stage type 2 diabetes mellitus rat model by Hernández-Martín M. et al. The article deals with the beneficial effect of dietary consumed silicon included in restructured meat (Si-RM) on dyslipidaemia associated with late-stage T2DM. The animal model is well established, experimental methods are highly relevant to the study aim, a bulk of different biochemical, histological, immunohistochemical and functional parameters were measured, the wide range of results was obtained, clearly presented and well discussed, all together strongly supporting the conclusion on mitigated diabetic dyslipidaemia with Si-RM.

I have few minor concerns:

Line 4: Delete full stop at the end of the title

Line 119: Add word „Feed“ before the word „intake“ at the beginning of the sentence

Line 274: Maybe to delete assumption: „Although, there was a tendency to improve”. Looking at the VLDLtg and HDLc values in Table 2, that improvement is not clear.

Table 3: Is there any need to put “a” next to the values in one row, which do not differ? Maybe the “NS” in last column is enough?

Line 539: Please check again, if ED needs to be substituted with “LD”? See Table 2

Kind regards

Author Response

Answer to Reviewer #1

Reviewer´s comments: I have read and revised manuscript foods-3028811, titled: Silicon as a meat functional ingredient improves jejunal and hepatic cholesterol homeostasis in a late-stage type 2 diabetes mellitus rat model by Hernández-Martín M. et al. The article deals with the beneficial effect of dietary consumed silicon included in restructured meat (Si-RM) on dyslipidemia associated with late-stage T2DM. The animal model is well established, experimental methods are highly relevant to the study aim, a bulk of different biochemical, histological, immunohistochemical and functional parameters were measured, the wide range of results was obtained, clearly presented and well discussed, all together strongly supporting the conclusion on mitigated diabetic dyslipidemia with Si-RM.

I have few minor concerns:

Comment 1- Line 4: Delete full stop at the end of the title

Reply 1: Thank you for your comment. The full stop in the title has been removed.

Comment 2- Line 119: Add word „Feed “before the word „intake“at the beginning of the sentence

Reply 2: Thank you for your comment. Following suggestion’s reviewer 2, we have rewritten the section 2.1. Diets, animal model and experimental design, from lines 99-139, so that sentence was also changed.

Comment 3- Line 274: Maybe to delete assumption: „Although, there was a tendency to improve”. Looking at the VLDLtg and HDLc values in Table 2, that improvement is not clear.

Reply 3: In reviewing table 2, we found your comment to be appropriate and the sentence has been deleted.

Comment 4- Table 3: Is there any need to put “a” next to the values in one row, which do not differ? Maybe the “NS” in last column is enough?

Reply 4: We agree that “NS” in the last column is sufficient. It was a mistake on our part to include the “a” in this case and in the new version it has been omitted. We thank you for the comment.

Comment 5- Line 539: Please check again, if ED needs to be substituted with “LD”? See Table 2

Reply 5: Thank you for pointing this out. We have rechecked the data, and indeed, it is a typo. We have replaced ED with LD in the new version of the manuscript and now it is in line 555.

Reviewer 2 Report

Comments and Suggestions for Authors

Comments: The article entitled “Silicon as a meat functional ingredient improves jejunal and hepatic cholesterol homeostasis in a late-stage type 2 diabetes mellitus rat model” evaluated the cholesterol-lowering effects and mechanism of restructured meat (RM) matrix (Si-RM) in a high-saturated-fat-high-cholesterol diet (HSFHCD) combined with a low dose of streptozotocin plus nicotinamide injection were used as late-stage type 2 diabetes mellitus (T2DM) model. This study is complete, well-designed, and conducted, but some questions need to be addressed. I suggest to polish and check it carefully.

1. Different cooking methods would affect the stable of silicon in restructured meat matrix? How it’s going when boiled, steamed, and fried.

2. Is silicon an accepted additives in food, do we have some policy and standard for this. What’s the content of silicon in restructured meat matrix, it’s safe?

3. I mentioned the authors didn’t explain the restructured meat matrix production processing, and what the form of silicon when adding to meat products.

Comments on the Quality of English Language

Minor editing of English language required

Author Response

Answer to Reviewer #2

Comments: The article entitled “Silicon as a meat functional ingredient improves jejunal and hepatic cholesterol homeostasis in a late-stage type 2 diabetes mellitus rat model” evaluated the cholesterol-lowering effects and mechanism of restructured meat (RM) matrix (Si-RM) in a high-saturated-fat-high-cholesterol diet (HSFHCD) combined with a low dose of streptozotocin plus nicotinamide injection were used as late-stage type 2 diabetes mellitus (T2DM) model. This study is complete, well-designed, and conducted, but some questions need to be addressed. I suggest to polish and check it carefully.

Comment 1. Different cooking methods would affect the stable of silicon in restructured meat matrix? How it’s going when boiled, steamed, and fried.

Reply 1: There is evidence that processing modifies meat properties and could also modify the bioactive ingredients included in functional meat (Gómez I, et al. Foods. 2020 Oct 7;9(10):1416). To avoid the interaction of culinary processing, in the present study, silicon has been included in unprocessed meat in order to know SI-RM hypocholesterolemic properties. In accordance with your suggestion and taking into account the interest of consumers, industry and public health, more studies would be necessary to investigate how different cooking systems (cooking, heating, boiling, frying, etc.) would affect bioavailability and function of silicon as a functional ingredient in processed meat. Our research group has a long history in the study and design of processed and unprocessed functional meats with different compounds used as bioactive ingredients. Therefore, your suggestion is viable, and we keep it in mind in the future in the medium and long term.

Comment 2. Is silicon an accepted additives in food, do we have some policy and standard for this. What’s the content of silicon in restructured meat matrix, it’s safe?

Reply 2:

Certainly, silicon is a food additive in the form of silicon dioxide (food additive code: E551). However, the silicon used in our study is choline-stabilised orthosilicic acid obtained from Silicium organique G57TM (Glycan Group, Genève, Switzerland) a soluble form which is used as a nutritional supplement.

In relation to the content of silicon in diet, in Table S1, you can find the silicon content in the experimental SI-RM diet.

The comments 2 and 3 indicate us that these concepts are not sufficiently clarified in the article, and we have modified section 2.1 Diets, animal model and experimental design, from lines 99-139, for further clarification.

Comment 3. I mentioned the authors didn’t explain the restructured meat matrix production processing, and what the form of silicon when adding to meat products.

Reply 3:

Following your suggestion and in order to clarify these concepts, we have rewritten the section 2.1. Diets, animal model and experimental design, from lines 99-139.

Restructured meat matrix (RM) was prepared according to Schultz-Moreira et al. [27]. Details of diet formulation and composition are in Supplementary Table S1. RM was processed from lean minced meat, consisting of an equal mix of pork and veal (50% each) from a local store (reference SF302; Safe Custom Diet, S.L. Augy, France). This mixture was blended with lard for one minute using a grinder-homogenizer connected to a cooling bath at 2°C (Stephan Universal Machine UM5, Stephan u. Söhne GmbH and Co., Hameln, Germany). After preparation, the meat mixtures were freeze-dried in a LyoAlfa 10 freeze dryer (Telstar, Terrassa, Spain) for 48 hours. The freeze-dried products were then ground into a fine, homogeneous powder using a refrigerated mincer (Stephan Universal Machine UM5, Stephan u. Söhne GmbH and Co., Hameln, Germany) for two minutes. Thus, each kilogram of diet consisted of 30% meat mixture and 70% purified formulated diet, which were mixed and sieved three times until a completely homogeneous powder was obtained. The silicon-enriched meat (Si-RM) was prepared in the same way as the control meat, but with the addition of silicon. The form of silicon used in the diet was choline-stabilised orthosilicic acid (H4SiO4) obtained from Silicium organique G57TM (Glycan Group, Genève, Switzerland), a major form of bioavailable silicon for both humans and animals. The silicon dose (2 mg/kg b.w./day) was based on the differences in silicon intake between Western and Eastern populations, as described by Garcimartin et al. [22]. To achieve the desired supplementation, a certain amount of organic silicon containing 67 mg of silicon was added to 1 kg of the meat mixture. This resulted in a final silicon concentration of 20 mg per kg of the overall diet which is safe as previous studies have demonstrated [25,26].

Twenty-four male Wistar rats, each two months old (Harlan S.L., Barcelona, Spain), were used in this study. The rats were randomly divided into three equal groups. After a seven-day acclimatization period, eight rats were fed a high-saturated-fat diet (HSFD) made of RM (50% pork/50% veal) for eight weeks to induce early-stage T2DM (ED). The other sixteen rats were fed a high-saturated-fat high-cholesterol diet (HSFHCD, HSFD with the addition of 1.4% cholesterol plus 0.2% cholic acid) based in RM for three weeks to induce late-stage T2DM (LD). Following this period, these rats received an intraperitoneal injection of streptozotocin (STZ, 65 mg/kg b.w.) and nicotinamide (NAD, 225 mg/kg b.w.) (both from Sigma Aldrich, Madrid, Spain). Four days later, fasting hyper-glycemia was confirmed, and the animals were divided into two groups: the LD group continued the RM/HSFHCD, while the LD-Si group received Si-RM/HSFHCD, with a Si dose of 2 mg/kg b.w./day. ED group represents early-stage T2DM characterized by insulin resistance and hyperglycemia with insulinemia, while LD group mimics a later phase of T2DM, characterized by higher glucose concentrations and hypoinsulinemia [25,26]. Rats had ad libitum access to tap water and food. Daily feed intake and body weight were recorded, and cholesterol intake calculated. During the final week, fecal excretion was measured daily. Rats were anesthetized with isoflurane (5% v/v) before euthanasia, and blood was collected from the descending aorta using a heparinized syringe. The jejunum and liver were dissected, weighed, and processed (Figure 1).